# A Reporting Quality Assessment of Systematic Reviews and Meta-Analyses in Sports Physical Therapy: A Review of Reviews

**DOI:** 10.3390/healthcare9101368

**Published:** 2021-10-14

**Authors:** Sung-Hyoun Cho, In-Soo Shin

**Affiliations:** 1Department of Physical Therapy, Nambu University, 23 Cheomdan Jungang-ro, Gwangsan-gu, Gwangju 62271, Korea; shcho@nambu.ac.kr; 2AI Convergence Education, Graduate School of Education, Dongguk University, 30, Pildong-ro 1 gil, Jung-gu, Seoul 04620, Korea

**Keywords:** assessment, meta-analysis, systematic review, physical therapy, review of reviews

## Abstract

This review of reviews aimed to evaluate the reporting quality of published systematic reviews and meta-analyses in the field of sports physical therapy using the Preferred Reporting Items for Systematic Reviews and Meta-Analyses (PRISMA) guidelines. This review of reviews included a literature search; in total, 2047 studies published between January 2015 and December 2020 in the top three journals related to sports physical therapy were screened. Among the 125 identified articles, 47 studies on sports physical therapy were included in the analysis (2 systematic reviews and 45 meta-analyses). There were several problems areas, including a lack of reporting for key components of the structured summary (10/47, 21.3%), protocol and registration (18/47, 38.3%), risk of bias in individual studies (28/47, 59.6%), risk of bias across studies (24/47, 51.1%), effect size and variance calculations (5/47, 10.6%), additional analyses (25/47, 53.2%), and funding (10/47, 21.3%). The quality of the reporting of systematic reviews and meta-analyses of studies on sports physical therapy was low to moderate. For better evidence-based practice in sports physical therapy, both authors and readers should examine assumptions in more detail, and report valid and adequate results. The PRISMA guideline should be used more extensively to improve reporting practices in sports physical therapy.

## 1. Introduction

Evidence-based practice (EBP) is the best decision-making method for patients [1]. The term “evidence-based medicine” was first introduced by Gordon Guyatt at McMaster University, in Canada in 1992 [2]. The Center of Evidence-Based Physiotherapy (CEBP), currently headquartered at the University of Sydney, Australia, was established in 1999 [3]. Clinicians should have the knowledge and skills to write and understand published systematic reviews and meta-analyses that will help better decision-making for patients [4]. Primary studies have several limitations in terms of clinical decision-making, e.g., a limited sample, mixed results, and inconsistent analytical methods and reporting [5]. Some scientific journals even tend to selectively publish thesis manuscripts with statistically significant results [6]. This may induce biases, such as result reporting bias, which may affect the validity of the results [4,7]. SRs and MAs overcome the limitations of primary studies [7].

Systematic reviews (SRs) and meta-analyses (Mas) are some of the best sources of evidence and scientific research, including clinical expertise and patient-reported outcomes [6]. The process of SR and MA includes problem formulation, literature search, data coding, and data analysis and reporting, which minimize the bias and increase transparency and reproducibility [8]. An SR is a literature review that collects, comprehensively analyzes, and evaluates all studies related to a research topic [9]. Therefore, if several clinical research papers show the same results through studies on the effect of the same treatment rather than a single clinical study, it provides a more accurate basis for the treatment effect than a single individual clinical study paper [10].

MA refers to a statistical method that synthesizes and analyzes quantitative research results extracted from multiple studies to produce summarized and organized empirical knowledge on a topic [7]. MA is a valid method for finding evidence so that clinicians and researchers can have a rationale for solving health-related problems [11]. Additionally, MA distinguishes the intervention effects of previous studies by characteristics and objectively compares them to develop intervention programs that can be successfully used in various practical fields [12].

SRs and MAs comprise the highest level in the evidence pyramid of medicine [13]. Systematic reviews (SRs) and meta-analyses (MAs) provide the best basis for use in decision-making regarding the application of medical service interventions [6,14]. Clinicians should have the knowledge and skills to write and understand published SRs and MAs, as this will aid in making better decisions with their patients [5].

The PRISMA (Preferred Reporting Items for Systematic Reviews and Meta-Analyses) guidelines detail what should be reported in MAs and SRs analyses [15]. However, the reporting quality of MAs and SRs in numerous clinical areas is low and very problematic [16,17,18]. Only one 2012 paper was found to analyze the reporting of SRs related to physical therapy, and it was also of very low quality [19].

Clinical intervention should be guided by scientific evidence, which should be obtained by valid and transparent processes and methods. Bias in the scientific process can lead to inappropriate clinical practices. Therefore, MAs and SRs should be conducted by referring to the PRISMA guidelines [20]. Quality assessment studies on MAs and RSs in other fields have been published. The purpose of this study is to indicate problems in research reporting methods and thus produce a more valid MA in sports physical therapy [20,21].

Interpreting the results of SRs and MAs should be performed carefully because the results may suffer from reporting study weakness [22]. There has been no quality reporting about SRs and MAs in the field of sports physical therapy research. Therefore, the purpose of this study was to evaluate the reporting quality of SRs and MAs published in the top three sports physical therapy journals [15]. This article summarized the major study results, the SRs and MAs method, and major findings and evaluated whether the reviewed articles aligned with the PRISMA guidelines’ 27 items.

## 2. Materials and Methods

### 2.1. Eligibility and Exclusion Criteria

#### 2.1.1. Eligibility Criteria

The top three journals in the field of sports physical therapy were selected based on the Journal Citation Reports impact factor (IF) index. The Journal of Orthopaedic & Sports Physical Therapy, Journal of Athletic Training, and Physical Therapy in Sports ranked first, second, and third in this specific category of journals with IFs of 3.839, 2.478, and 1.926, respectively. The IF index measures the average number of citations received in a particular year by papers published in the journal during the preceding years of 2019 and 2020 [23]. The article selection criteria were as follows: (1) peer-reviewed articles published between January 2015 and December 2020, and (2) a statistical MA or systematic literature review of an intervention program.

#### 2.1.2. Exclusion Criteria

As this study was not meant for clinical effect analysis, only treatment effect SRs from the three top journals were targeted. Narrative reviews, diagnostic test SRs, primary studies, qualitative review articles, authors’ opinions, letters, and abstract presentations were excluded.

### 2.2. Information Sources and Search Strategy

#### 2.2.1. Electronic Search

We searched the Journal of Orthopaedic & Sports Physical Therapy (https://www.jospt.org, accessed on 31 April 2021), Journal of Athletic Training (https://natajournals.org/loi/attr, accessed on 31 April 2021), and Physical Therapy in Sports (https://www.journals.elsevier.com/physical-therapy-in-sport, accessed on 31 April 2021). The search strategy utilized a combination of medical subject heading (MeSH) terms and free text words, including “meta-analysis” (MeSH), “meta-analysis” (text word), “review” (MeSH), and “systematic review” (text words).

#### 2.2.2. Manual Search

The included studies returned by the search, and previously published SRs and MAs related to the topic, were screened to identify any additional studies which could fit the criteria.

### 2.3. Study Selection

Publication details of all studies identified in the literature search were exported to EndNote (Endnote X9.3.3, Clarivate Analytics, Philadelphia, PA, USA). Once all records were imported, duplicates were removed. After that, titles and abstracts were independently screened for eligibility by the two authors using the specified inclusion and exclusion criteria. Full texts of the remaining articles were then sourced and independently evaluated for inclusion. Any disagreement was resolved through discussion.

### 2.4. Data Collection Process and Data Item

The PRISMA statement includes a checklist of 27 items [21], and the PRISMA checklist was used by the raters in the analyses of the eligible SRs [21]. For each checklist item, it was established that a rating of “yes” would only be assigned if the PRISMA statement recommendations were fully complied with. If the rater considered the information regarding any item to be incomplete, missing, or doubtful, that item was rated as “no.” The quality evaluation criteria of this paper were identified as follows: “low quality,” “moderate quality,” and “high quality” when there was <50%, <75%, and >75% agreement with the PRISMA checklist items, respectively [19,22,24,25,26].

The characteristics of the participants and interventions in the 47 selected articles were identified and coded per the research evaluation framework (Appendix A) [19]. The articles were listed and outlined according to the coding table (Appendix B).

The two authors also independently extracted the following data from each included article into predesigned coding sheets: (1) study identification: the first author’s name, location of the corresponding author(s), year of publication, and journal name; (2) number and design of the studies in the MA/SR; (3) population (participants); (4) interventions; (5) comparison between interventions; and (6) outcome measures. Discrepancies were resolved through discussion.

### 2.5. Reliability of the Evaluators

Two researchers independently checked each checklist item in a coding sheet and resolved discrepancies by consensus via weekly Zoom meetings. If the disagreement was not resolved, the researchers planned to consult a third coder to make the final decision. However, there were no disagreements between the coders in any of the items. Some items were not fulfilled according to the PRISMA checklist. The two researchers also evaluated additional reporting and statistical issues based on the PRISMA 2020 statement [27].

### 2.6. Planned Methods of Analysis

#### 2.6.1. Reporting of Epidemiological and Descriptive Characteristics

The epidemiological and descriptive characteristics of the included MAs and SRs were assessed according to the journal type, corresponding author’s location, number and design of included studies, population/patients/defects, type of intervention, comparison between interventions, outcome (Appendix B).

#### 2.6.2. Statistical Analysis

Each PRISMA checklist item was presented as either the ratio or percentage of how many of the 47 PRISMA articles were properly followed (Appendix C).

### 2.7. Ethical Statement

This study was approved by the institutional review board of Nambu University (ethical code: 1041478-2017-HR-016; approval date: 8 November 2017) and was conducted in accordance with the ethical standards of the Declaration of Helsinki.

## 3. Results

### 3.1. Study Selection

A total of 2047 studies published by the Journal of Orthopaedic & Sports Physical Therapy, Journal of Athletic Training, and Physical Therapy in Sports between January 2015 and December 2020 were considered. Among these, 47 articles listed as a systematic literature review or MA in the article category were initially selected. We excluded five narrative review articles, two articles that reported a rate, three articles that reported a ratio, two articles that reported the prevalence, one article that reported intraclass correlation coefficients, one article that reported reliability and validity, and one article that presented the diagnostic accuracy of a clinical test. Consequently, 47 articles were included in the final analysis (Appendix D) (Figure 1).

### 3.2. Study Characteristics

#### 3.2.1. Epidemiological and Descriptive Characteristics

MAs and SRs published in the Journal of Athletic Training, Journal of Orthopaedic & Sports Physical Therapy, or Physical Therapy in Sports are shown in Appendix B. The types of interventions varied across the wide-ranging sports physical therapy field. The reporting guidelines used for the SR process also varied: 32 (68.1%) articles used the PRISMA guidelines [15]. However, 15 (31.9%) articles did not describe the reporting guidelines used. Regarding funding sources, 10 studies received private and/or public support (21.3%), and 37 studies received no funding.

#### 3.2.2. General Characteristics of the Included Studies

The main characteristics of all the included studies are described in Appendix B. The reporting quality of key components of the MAs and SRs, as evaluated based on the PRISMA guidelines, is shown in Appendix C. Two papers performed an SR, and the remaining 45 conducted a MA. Furthermore, 27 (57.4%) studies included an RCT. Forty-two (89.4%) studies used outcomes that were continuous variables, such as the mean change difference.

### 3.3. Synthesis of the Results

The evaluation results are described below in the following order: title, abstract, introduction, methods, results, and discussion (Appendix C).

#### Reporting of the General Components of the SR Process (27 Items)

To enhance the validity and impact of SRs, all authors and editors must apply established reporting standards. Thirty-two articles mentioned the application of a guideline and 15 articles did not. Thirty-two studies adhered to the PRISMA guidelines and two studies applied the 2015 PRISMA-P (Preferred Reporting Items for Systematic Reviews and Meta-analysis Protocols) guidelines [28,29]. One of the studies [28] mentioned the Measurement Tool to Assess Meta-analysis of Observational Studies in Epidemiology (MOOSE) [30].

Regarding item 2 (structured summary), we checked whether the abstract specifically presented the analytical model and effect size. Powden et al. specified that a fixed or randomized model was used [31]. Furthermore, studies that only reported the odds ratio (OR), relative risk (95% confidence interval [CI]), and effect size (Cohen’s d, 95% CI) in the data synthesis methods, and studies that did not present specific results (numbers) in the abstract, were considered to not have a structured summary (item 2). The PRISMA summary guidelines are a better source for this item published in 2013 [20].

In primary studies, researchers tend to report statistically significant results and emphasize a positive and large effect [32]. SRs and MAs have similar tendencies and, thus, require protocol registration to prevent selective reporting. Protocol registration is one of the methods that increase the validity of a MA [9]. Therefore, the protocols of SRs and MAs should be registered in PROSPERO (an international database of prospectively registered SRs in health and social care by the University of York, which is accessible to the public and researchers) [33], as for primary studies. However, no studies reported that the protocol was registered in PROSPERO or a local research foundation (item 5). For protocol and registration (item 5), none of the studies reported a selection/reporting bias.

The registration rate in the medical field is also very low, at 21% [34]. Protocol registration should be emphasized because selective reporting can be evaluated by comparing the protocol against the full paper [34,35]. The National Institute for Health Research (NIHR) also supports protocol registration [36].

Twenty-eight (59.6%) studies did not report the methods used to assess the risk of bias of studies (item 12). Furthermore, 23 (48.9%) studies did not report the methods used to assess the risk of bias across studies (e.g., publication bias) (item 15), and 28 (59.6%) studies did not describe the methods of additional analyses (item 16). Furthermore, more than half of the reviews did not report the “risk of bias within studies” (data presented on the risk of bias of each study and, if available, any outcome level assessment). This finding was also consistent with previous studies [22]. Reporting bias is crucial for assessing effect size because only advantageous results may be reported [37].

In the results section, 19 (40.4%) studies did not present data on the risk of bias for each study and any outcome level assessment, if available (item 19), and 23 (48.9%) did not report the risk of bias assessment across studies (item 22). Heterogeneity means the degree of differences in the results of each single research finding [38]. Through MAs, scholars calculate the heterogeneity index to understand the primary factors that impact individual studies’ effect sizes [39]. For heterogeneity tests, Q (34/47) and I^2^ inconsistency (38/47) statistics were used. Reporting on the effect size, most studies (42/47) did not provide the effect size formula, and no study reported the effect size variance formula in their methods section [40].

In this review, very few studies reported independent assumptions, missing data, or outliers, which should be addressed. Independent assumptions comprise two issues: first, whether the same sample was used twice or not, and second, how more than one effect size was treated in calculating an effect size for MA [40].

The results of additional analyses were mentioned in only 25 (53.2%) studies (e.g., sensitivity or subgroup analyses and meta-regression analysis) (item 23). Forty-two articles did not describe a sensitivity analysis. The sensitivity analysis, used to test the reliability of the cumulative effect across included studies, revealed the effect sizes [41].

Interestingly, one study [42] evaluated the reviewed articles using the Grading of Recommendations Assessment, Development and Evaluation (GRADE) working group criteria after they completed their MA [43,44]. GRADE was used to investigate the overall quality of evidence for each outcome [45]. The MA evaluated the study’s design, risk of bias and publication bias, consistency, the complexity of interventions, and roughness [46]. The Cochrane Handbook for Systematic Reviews of Interventions also supports the evaluation process by researchers who completed the reviews [14]. The GRADE criteria include five quality evaluations: (a) risk of bias, (b) inconsistency, (c) indirectness, (d) imprecision, and (e) publication bias [47].

In the discussion section, comments about the study limitations, outcome level, and review levels, such as the risk of bias and incomplete retrieval of identified research and funding, were not provided in 37 (78.7%) studies (item 27). Many studies did not report a funding source. Research results can fundamentally differ according to funding sources. Funding bias refers to when a study’s outcome is more likely to support the interests of the organization funding the study [48]. For example, studies regarding omega-3 supplementation for the prevention of cardiovascular disease, or the relationship between cell phone use and the risk of brain tumors, showed contrasting results depending on the funding source [49,50]. Most research indicates that sucralose is safe—except for research sponsored by competitors [51].

## 4. Discussion

This study was benchmarked against previous studies, which were mainly published in top journals according to the IF criteria related to the reporting quality assessment [52,53,54,55]. The trends of the last 5 years were assessed because the PRISMA-Diagnostic Test Accuracy, PRISMA-Rapid Reviews, PRISMA-Scoping Reviews, and PRISMA-Network MA guidelines were released in 2015. Thus, the current year was when systematic literature review and MA reporting standards began to be subdivided [56]. This review evaluated 47 MAs and SRs reported in the Journal of Orthopaedic & Sports Physical Therapy, Journal of Athletic Training, and Physical Therapy in Sports related to sports physical therapy, using the PRISMA guidelines.

The study results were very similar compared to analyses of reporting of systematic reviews in physical therapy [19]. Analysis of reporting of systematic reviews in studies of reporting standards using the PRISMA statement have reported information regarding the risk of bias, protocol and registration, additional analysis, and funding was insufficient in quality-evaluated studies in the field of nursing [24], acupuncture [25], and diagnostic testing [26]. In the field of sports physical therapy, protocol registrations (38.3%), risk of bias across studies (51.1%), additional analysis (40.4%), and funding (21.3%) were the most problematic PRISMA items. Although the quality of SR and MA reporting in sports physical therapy was medium-to-low, similar to that in other clinical fields, key reporting components of the SR process were missing in most of the MAs and SRs. The critical appraisal of such studies must improve these reporting issues, which pertain to general MAs (27 items). For better EBP in sports physical therapy, authors and readers should examine assumptions in more detail, and report valid and adequate results. The PRISMA guidelines should be used more extensively to improve reporting practices in physical therapy.

Protocol registration is increasingly recommended in clinical trials [57] and SRs [34], but this study showed a low protocol enrollment of 38.3% (item 5). In a previous survey, only about one-fifth of SRs in physical therapy were registered, indicating that the enrollment rate was low [58]. According to the Cochrane Handbook for systematic literature review of interventions, the prospective registration of the protocol reduces the author’s bias by publicly documenting a priori planned methodology [34,36]. Importantly, the registered SRs showed significantly higher methodological quality compared to the unregistered SRs. The protocol provides transparency and clarifies the hypothesis, methodology, and analysis of the SRs and MAs undertaken.

Quality assessment of meta-analysis for randomized controlled trials (RCT) was performed using the Risk of Bias tool developed by the Cochrane group [59]. The NRS tools for Newcastle and Ottawa Scale (NOS) [60], and Risk of Bias Assessment tool for Non-randomized Studies (ROBANS) [61]. However, there may be some confusion in the concepts of the reporting standards for individual studies (CONSORT statement) [62]. Strengthening the Reporting of Observational Studies in Epidemiology (STROBE) statement [63], and quality assessment tools for meta-analysis (ROB, NOBANS) [64].

One study reported the STROBE statement as a reporting standard for individual studies, as opposed to a quality assessment tool for meta-analysis [65]. The STROBE statement is a checklist of items that should be included in reports of cohort studies. An explanation and elaboration article discussed each checklist item and provided the methodological background and published examples of transparent reporting [63]. The reporting criterion for meta-analysis of observational studies should be evaluated as MOOSE [30].

For summary measures, the type of effect size measure used in the manuscript must be described. The OR and standardized mean difference (SMD) as summary measures were indicated in the studies (item 13). The effect size is a key concept in a meta-analysis, and an essential component of quantitative research reporting and hypothesis testing [40]. In many studies, a corrected SMD, i.e., Hedges’ g with 95% CI was computed in consideration of the small sample size, and the inverse of the variance was used as the weight for each effect size [66]. As shown herein, studies need to classify the effect size computation, variance computation, and equations, and present them in an easily comprehensible manner for readers. However, only 5 of 47 studies described the effect size computation [29,65,67,68,69] (Appendix A). One benefit of evaluating the effect size is that it quantifies the difference between groups in the observed data [70]. Moreover, the effect size is presented as a standard deviation; thus, it can be compared between studies and utilized in MAs, as well [71]. Researchers in the field of clinical medicine should also recognize the benefits of the effect size and use it widely in medical research.

Subgroup analyses further decrease the number of studies and thus weaken the power of the analysis, necessitating a careful interpretation of the data. Additionally, only 1 of the 23 studies performed a multivariate analysis [72].

The quality evaluation level of the 47 studies that were investigated revealed low to moderate and very low levels in statistical issues, such as mentioning the effect size formula and variance in the methods section. MAs use a summary measure with a statistically known variance [73], and the effect size variance formula is related to the distributional assumption such as the normality and homogeneity assumption for hypothesis testing. The results of effect size computation (like the SMD dealing with continuous variables, correlations, and odds ratios with dichotomous variables) should be provided for each individual MA study as major characteristics of the included studies [27,40].

There were some flaws in the reviewed articles, and some suggestions were proposed for more valid MA results. Applying appropriate MA assessment tools for physical therapy research is required to increase the reviewed articles’ validity. The synthesis of research studies in physical therapy research includes various study designs such as RCTs, observational studies, and scholars must consider the appropriate method of exploring validated reviews in physical therapy research.

As of 2021, version 6.0 of the Cochrane Handbook for Systematic Reviews of Interventions has been released [14]. The ROB 2 is the gold standard to evaluate the quality of RCT biases [39] and is a reorganized form of the Cochrane ROB tool by the same team [74]. The key features are that researchers can simply decide if a bias exists in the reviewed research and can evaluate bias for particular result findings within an RCT design and beyond the RCT design [75]. The Risk of Bias in Non-randomized Studies of Interventions (ROBINS-I) was also developed to evaluate intervention studies with nonrandomization in the Cochrane handbook (version 6.0). The main differences are that reviewers can easily critique the bias (using “low,” “moderate,” “serious,” and “critical”), focusing on the after-intervention effects [76].

Researchers should use appropriate, updated quality assessment tools to reflect a study’s research design and objectives and consider the complexity of interventions, proper groupings, and scientific effect size calculations. In addition, the present study proposed measures to improve the quality of MAs and ultimately aimed to examine the current situation, contributing to the enhancement of the EBP of sports physical therapy.

### 4.1. Limitations

To our knowledge, no study has investigated the reporting quality of SRs and MAs in the field of sports physical therapy. However, the present study has some limitations.

The database searches were only conducted in three specific journals based on their IF in the field of sports physical therapy. We need to comprehensively search the relevant databases related to sports physical therapy to conduct future SRs. For example, other journals from the field of sports and exercise medicine (e.g., British Journal of Sports Medicine) are important and should be included in the next study. We will extend the more studies to for 10 years in the next study.

The authors of the reviewed articles may have used the appropriate method but omitted important details from the report or removed key information during the publication process. We were in the position of the reader and could evaluate what was reported in the articles only. Additionally, there may be limitations in that the scope of the research may be different from the part where the actual research was conducted because the researcher reported based on the writing of a paper.

Quality thresholds were based on previous studies and were not agreed to be absolutely not interpreted. In addition, we determined that the quality of the performance was appropriate only if the reporting was adequate in terms of each checklist. If a comprehensive quality evaluation of the reporting criteria is conducted by including studies other than those in the top journals, the issue of low quality could be even more serious.

Because of research ethics, sports physical therapy studies need to perform general physical therapy in addition to the major intervention; thus, the lack of studies that only examined the physical therapy intervention poses a considerable limitation. In the future, when conducting systematic literature reviews and MAs, it will be helpful to improve the quality of clinical studies only when non-random studies, as well as RCT studies, are included due to the nature of clinical studies in physical therapy.

### 4.2. Clinical Implications

The findings of our study suggest that physical therapy studies need to be designed appropriately as per the purpose of the study and that physical therapy programs for patients should be structured more systematically. Considering the lack of previous studies that qualitatively evaluate and emphasize clinical judgment, future studies need to discuss complex interventions and network MAs as recent research trends.

## 5. Conclusions

This critical assessment demonstrated that the current quality of reporting and conducting of MAs and SRs is low to moderate, as it is in other medical disciplines. Problem areas of current meta-analyses and SRs include the exploration of the risk of bias across studies, protocol registrations, and additional analyses. Performing a meta-analysis with inadequate reporting increases the risk of invalid results in a meta-analysis.

Therefore, a reporting guideline, such as the PRISMA statement, is helpful for authors when writing meta-analyses and SR.s Clinicians need to have a thorough knowledge of research methodology, to render the interpretation of sophisticated statistical analyses easier. In addition, the present study proposed measures to improve the quality of meta-analyses, and ultimately aimed to examine the current situation, contributing to the enhancement of the practice of sports physical therapy.

## Figures and Tables

**Figure 1 healthcare-09-01368-f001:**
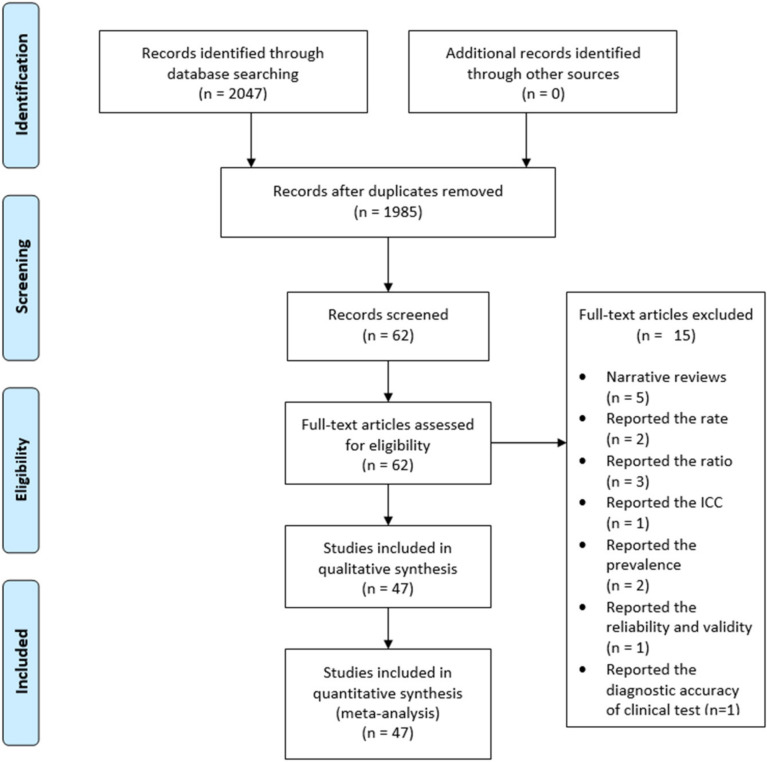
Flow chart describing the selection of the 47 articles after multiple phases of the screening process. ICC, intra-class correlation.

## Data Availability

All data relevant to the study are included in the article. Data were collected from studies published online or publicly available, and specific details related to the data will be made available upon request.

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
