# Peer review of "A Reporting Quality Assessment of Systematic Reviews and Meta-Analyses in Sports Physical Therapy: A Review of Reviews"

_healthcare, 2021, doi:10.3390/healthcare9101368_

Round 1
Reviewer 1 Report
The paper analyze 47 papera, and pretend to draw conclusions on whole sport phyisicla theraphy reviewes, irrespective of type of injury, age, sport, localization of the injury, grade of the injuries and so on. This appear a little bit over-ambitious.
The authors claims that no article exist on review on reviews in PT and this paper is the first one. Probably there is a reason. As I stated above, is quite impossible due to the variability of the possible conditions.
Albeit the paper is statistically sound in the structure, it starts on unstable basis.
Also it makes weird assumption about a possible funding bias citing only 2 papers. This statement question (based on 2 papers) the scientific integrity of authors. This seems at least weak.
In summary, draw conclusion on the weakness of reviews in the whole PT, irrespective of type of injuries, age and sex, location, sports and degree of the injuries, is at least imaginific.
In summary, to use methods who has originally developed for reviews, and use it for reviews of reviews, introduce significants biases, that could not be accepted.
Author Response
We attached the resonse file.

Reviewer 2 Report
i found the approach to be one needed for each unique aspect of health care. It would be helpful if the Introduction was a bit more extensive with regard to perhaps a bit of history of systematic reviews, meta-analyses. The identification of the concerns as well as strengths of the processes was and is interesting and definitely important for the professionals who often depend on such to support or suggest approaches for treatment.
on lines 95-98, indicate the rationale for use of quality determination and labeling. Wondering if there is sufficient reasoning for the yes/no only response. In addition, a more thorough explanation regarding PRISMA in its various stages and detail would be helpful.
Very interesting that the review of the reviews was bold and accurate enough to present areas in need of attention.
Some specificity with regard to remarks: 2.1.2 offered good clarity of limitation to the journal sources.; in 3.2 - is there reason to expect a difference in the review process if types of research as well as number of subjects is considered independently?; The Appendices were nicely done and clearly delineated for reader.
likely it will be advantageous to pursue in the future and in detail the kinds of or degree of bias identified, avoided or perhaps expected.
Author Response
We attached the resonse file.

Reviewer 3 Report
- The authors studied A Reporting Quality Assessment of Systematic Reviews and Meta-Analyses in Sports Physical Therapy: A Review of Reviews. This study is informative, but some issues need to be fixed.
Introduction
- Lines 27-30: This first paragraph is incomplete, and also it needs to include more references for better support.
- Line 31: Before discussing evidence-based practice as the best decision-making method, You need to briefly list and discuss the other methods.
- Lines 36-37: Provide more references to support the following sentence.
“Primary studies have several limitations in terms of clinical decision-making, e.g., a 36 limited sample, mixed results, and inconsistent analytical methods and reporting”
- Lines 37-38: Provide more references to support the following sentence:
“SRs 37 and MAs overcome the limitations of primary studies”
Discussion
- The discussion section of this manuscript is the biggest concern. It needs to be way more elaborated and be described more comprehensively to get ready for publication.
Author Response
We attached the response file.

Reviewer 4 Report
1. In general, I suggest some extensive revision to the introduction for clarity and better definitions of ideas to justify the need for this review. For example:
- The first sentence (line 27-29) – not sure if you are referring to physical therapy research or physical therapy as a profession (eg., clinically).
- Similarly, with the second sentence (line 29-30) – this is very unclear, synthesized what results? What findings?
2. Lines 43-45 – I am not sure the argument that since PRISMA has been around for 10 years it is a good time to evaluate the quality of reviews in physical therapy research is sufficient. This also does not flow with the next sentence (lines 45-46) which states that improvement is still needed. Improvement in what?
3. Please provide justification for including only the top 3 IF journals (lines 58-60). Why were other journals from the field of sports and exercise medicine (e.g., British Journal of Sports Medicine) not included?
4. Please provide justification for searching articles published in the last 5 years only (lines 64-65).
5. How was the search conducted (lines 76-79)? Did you search systematically through the journal website or use another database?
6. More justification is needed for the establishment of quality thresholds (lines 94-97).
7. More detail on the screening criteria and process is needed (lines 134-137). What database was searched? Using what keywords? How might someone replicate this search?
8. Lines 256 – 263 – I am not sure this belongs in the discussion section.
9. Line 264 – what results? What previous study? The discussion section requires extensive revision and reference back to the original focus of this review.
Author Response
We attached the response file.

Reviewer 5 Report
This submission reported the quality assessment of systematic reviews and Meta-Analyses in sports physical therapy. This submission is well organized. A revision is suggested.
- According to the findings from this paper, please suggest the guidelines for future sports PT studies.
- This study screened studies published between January 2015 and December 2020 in the top 3 journals related to sports physical therapy. In my view, 10 years screening is suggested. This paper should include more studies and make a further conclusion.
- How to justify the top 3 IF journals, again, this paper should include more studies and make a further conclusion not just include top 3 IF journals.
- Please address the limitation of this study, please focus on the methods used in this study.
Author Response
We attached the response file.

Reviewer 6 Report
Thank you for allowing me to review the manuscript entitled "A Reporting Quality Assessment of Systematic Reviews and Meta-Analyses in Sports Physical Therapy: A Review of Reviews".
The purpose of this study was to evaluate the reporting quality of Systematic Reviews and Meta-Analyses published in the top 3 sports physical therapy journals and significant findings and evaluated whether the reviewed articles aligned with the PRISMA guidelines' 27 items.
This study is well structured and easy to read and understand. So I will only suggest a few small changes for improvement and some minor errors.
In the material and methods section, reference is made to the IF of the three selected journals. It would be appropriate to add the year in which this impact factor was chosen.
The table in Appendix A would improve readability if it were on a horizontal page.
In the table in Appendix B, there are 100.0% that should be changed to 100%, like the rest of the table.
I believe that part of the conclusion should be carried over into the discussion or rephrased. It is not clear whether this is a recommendation or an extract from the study itself. Line 339-343
Author Response
We attached the response file.

Round 2
Reviewer 1 Report
The manuscript has been significantly improved. In the present form it is highly acceptable for the publication.
Author Response
Thank you for reviewer
Reviewer 4 Report
The authors have addressed my concerns.
Author Response
Thank you for reviewer
This manuscript is a resubmission of an earlier submission. The following is a list of the peer review reports and author responses from that submission.
Round 1
Reviewer 1 Report
Firstly, I would like to thank the authors for their recent submission regarding methodological quality of systematic reviews in sports physical therapy. Methodological reporting is an important issue for all areas and despite guidelines being consistently developed and updated, issues remain. Therefore, I see the value in the work being completed and the results show the disparities around reporting in the field. I do have some concerns that I have highlighted below:
Major concerns:
The database searches are only conducted in three specific journals, based on impact factor the authors suggest these to be the top three journals in the area. No rationale is provided for such specificity of searching, other than the impact factors, which could add bias to the review. Why did the authors not search databases that would be relevant to the area of sports physical therapy (e.g. CINAHL, Sport Discus, and PubMed)? The journals mentioned are indexed in these databases. I think this needs to be made clear that it is a limitation of the study, with appropriate justifications. Additionally, no rationale is provided for restricting the year of publication to 2015.
Minor amendments:
Line 83 – two authors screened the articles independently to remove duplicates, seems rather odd. Did the authors use any software to conduct screening? Software such as endnote would automatically remove duplicates. If this was the case, please include this information here.
Line 93 – “coded as per the research evaluation framework”. Needs a reference and some more detail as to how this works, could be supplemental. As it may be unclear to someone not familiar with the framework as to how the studies have been ranked.
Line 108 – first mention of the use of PRISMA criteria, I would include this in the above section alongside the REF coding you have completed. As it currently stands it is a little disjointed and could be clearer.
Line 112 – change “Mas” to “MAs”
Line 127 – Appears to be a slight issue with formatting of the flow chart overlapping the writing.
Author Response
I attached the word file.

Reviewer 2 Report
The authors undertook a research to evaluate the methodological quality of systematic reviews in the field of sports physical therapy, which is welcome, as the outcomes of this study may help improve the quality of future systematic reviews in this specific domain. However, there are some major flaws in this manuscript, that need to be corrected before it can be considered for publication. Here are some comments/suggestions and questions to help improve the quality of this research and of its reporting.
- In the abstract, the authors stated that the review “aimed to evaluate the methodological quality of systematic reviews and meta-analyses in the field of sports physical therapy physical rehabilitation, using the Preferred Reporting Items for Systematic Reviews and Meta-Analyses (PRISMA) guidelines”. Please, note that PRISMA is not a methodological quality assessment tool, as it can be read on the PRISMA website: “PRISMA may also be useful for critical appraisal of published systematic reviews, although it is not a quality assessment instrument to gauge the quality of a systematic review” (http://www.prisma-statement.org/).
- In the methods section of the abstract, the authors identified the study as a “scoping review”. Please, note that this is not a scoping review, but rather an “umbrella review” or a “review of reviews”.
- Why has the literature search been limited to “January 2015 and December 2020 in the top 3 journals related to sports physical therapy”? This needs to be clearly explained in the full manuscript (methods section).
- It seems that the authors have confused the “quality of reporting” and the “methodological quality” of systematic reviews, as it can be easily seen through the conclusion of the study (abstract): “The quality of the reporting of systematic reviews and meta-analyses was low to moderate, as is found in other medical disciplines. The appropriate guidelines should be used more extensively in order to improve the reporting practices in the field of sports physical therapy”. Indeed, the title of the paper, as the aim statement, rather refers to the “methodological quality” of the systematic reviews: “A Methodological Quality Assessment of Systematic Reviews and Meta-Analyses in Sports Physical Therapy”. This confusion also appears in all the other sections of the manuscript, and should be corrected.
- All the comments made above, based on the abstract, should also be strictly taken into account for correction of the introduction, the methods, the results, the discussion and the conclusion sections of the full manuscript. For more information regarding the critical difference between the “methodological quality” and the “completeness of reporting” of systematic reviews, the authors may refer to the paper by Pussegoda et al. (2017) (DOI 10.1186/s13643-017-0527-2). Therefore, depending on the focus of this research, as should be clarified by the authors, they should make the choice of the appropriate assessment tool.
- Please note that a reporting guideline for review of reviews is currently under development (PRIOR, Preferred Reporting Items for Overviews of Reviews), which is different from classical PRISMA extensions (doi: 1186/s13643-019-1252-9). However, a reporting guideline by another group (The Joanna Briggs Institute,) is currently available, that needs to be used for the reporting of this research. The DOI of the paper on this guideline is: DOI: 10.1097/XEB.0000000000000055. Of course, the PRISMA guidelines for systematic reviews may also be of valuable help for reporting this research, even if this is not specific to this type of research.
- In the methods section, page 2, the authors listed the following as criterion for selection of articles: “the effect size was reported”. How could “effect size” be reported in systematic reviews not involving meta-analysis, as systematic reviews were also planned be included in this research? Please, revise.
- Please note that exclusion criteria are not opposite of inclusion criteria. Then, revise the exclusion criteria as reported in the current version of the manuscript.
- Please note that a full and detailed search strategy for at least one bibliographic database (e.g., Medline, Embase, Scopus, etc.) should be provided with the manuscript, for publication. In order to avoid delaying publication of this paper, the reviewer strongly advises the authors to ask for the help of a researcher with good experience in building search strategies for systematic reviews. The specific databases searched should be reported in the abstract as well as in the methods section of the full paper.
- The authors reported in their conclusion (abstract) that “the quality of the reporting of systematic reviews and meta-analyses was low to moderate”. How did the authors rate the quality of reporting as “low” or “moderate”, using the PRISMA guideline?
- The authors reported in the results section of the full manuscript that “None of the studies mentioned the Measurement Tool to Assess Systematic Reviews (AMSTAR). One of the studies [21] mentioned the Measurement Tool to Assess Meta-analysis of Observational Studies in Epidemiology (MOOSE) [23].” The reviewer is quite amazed to read this. Why should a systematic review of literature mention AMSTAR or MOOSE?
- Please note that the discussion section should be importantly amended. Indeed, some statements have nothing to do with what has been assessed in this research, or are made in a confusing
- The reviewer strongly advises the authors to take the time to fully revise this paper, according to the comments made here, before resubmitting the manuscript to the Journal.
Author Response
I attached the word file.

Round 2
Reviewer 2 Report
The reviewer thanks the authors for their careful consideration of comments raised during the first peer-review round. Here are some additional comments.
Please note that the fact that “evaluation based on top journals” has been undertaken in other fields or by other researchers should not justify the use of such approach as a standard. The “the impact factor criteria” is not (necessarily) a good choice. The reviewer is particularly not at all keen of this, and would advise the authors to consider comprehensively searching the relevant databases for future systematic reviews, as adequately suggested by reviewer 1.
What is the meaning of this: “Depending on the focus of the study by Pussegoda et al. [25], as should be clarified by the authors, we should make the choice of the appropriate assessment tool.”?
Please note that the “Data collection process and data items” section should refer ONLY to the data collection process, as well as to data items. Also, the “Eligibility criteria” section should describe ONLY the eligibility criteria.
Regarding the rating of the quality of reporting as “low” or “moderate”, the reviewer would like to draw that authors’ attention on the fact that having some previous articles mentioning this does not mean that this is something correct or acceptable. On which basis do the authors rate the quality of reporting as “low” or “moderate”? Which criteria were used to differentiate “low” from “moderate” quality?
Sorry for the comment about MOOSE, but please note that AMSTAR is a measurement tool to assess the methodological quality of systematic reviews (https://amstar.ca/Publications.php). In the view of the reviewer, there is no need that a systematic review of randomised or non-randomised studies mention AMSTAR, even if mentioning it may be understandable.
Please, strictly refer to the adequate reporting guidelines to improve the reporting of the methods section of this paper. Thank you.
Author Response
I attached the "response to reviewer" file.
